**Data Availability Statement:** The individual pseudonymized data, as used for the analysis, would allow some institutions (e.g. health

# Perceived empowerment and the impact of negative effects of the COVID-19 pandemic on the quality of life of persons with severe mental illness

Annabel Sandra Mueller-Stierlin[1,2]*, Friedrich Meixner[1], Jutta Lehle[1], Anne Kohlmann[1], Mara Schumacher[1], Stefanie Woehler[1], Anke Haensel[1], Sabrina Reuter[1], Katrin Herder[1], Nicole Bias[1], Thomas Becker[3], Reinhold Kilian[1]

1 Department of Psychiatry II, Ulm University, Günzburg, Germany, 2 Department of General Practice and Primary Care, Ulm University, Ulm, Germany, 3 Department of Psychiatry, University of Leipzig, Leipzig, Germany

* annabel.mueller-stierlin@uni-ulm.de

## Abstract

### Purpose

Beyond its direct effects on physical health the COVID-19 pandemic has been shown to have negative effects on the living situation of people with severe mental illness (SMI). To date, there has been little research on resilience factors preventing people with SMI from experiencing negative effects of the COVID-19 pandemic.

The objective of this study was to investigate the role of perceived empowerment (PE) as a resilience factor, preventing people with SMI from experiencing negative effects of the COVID-19 pandemic on daily living.

### Methods

We investigated negative effects of the COVID-19 pandemic on daily living in 931 persons with SMI at two times within six month between June 2020 and Mai 2021. To take into account the longitudinal structure of the data we applied mixed effects regression analyses and longitudinal path models.

### Results

A majority of participants experienced negative effects of the COVID-19 pandemic on several dimensions of daily living. Negative effects increased with rising levels of illness-related impairment but decreased as the level of PE rose. While negative effects of the COVID-19 pandemic at follow-up were negatively associated with overall subjective quality of life baseline, PE was negatively associated with the negative impact of the pandemic and positively with quality of life.

insurance providers and service providers involved in the project) to identify the respective participants based on their routine documentation. Though, the datasets generated during and analysed during the current study are available from the Department of Psychiatry II, Ulm University (psychiatrie@bkh-guenzburg.de) on reasonable request. The project acronym "GBV" should be indicated along the request.

**Funding:** This study and most authors (AMS, FM, JL, AK, MS, SW, AH, SR, KH, and NB) are funded by the Innovation Fund from the Federal Joint Committee (G-BA, https://innovationsfonds.g-ba.de/), grant number 01NVF18028 (GBV). The funder was not involved in the study design, data collection and analysis, decision to publish, or preparation of the manuscript.

**Competing interests:** The authors have declared that no competing interests exist.

## Conclusion

Patients with SMI need support to reduce negative effects of the COVID-19 pandemic on their quality of life. The promotion of PE could help strengthen resilience in this target group.

## Trial registration

German Clinical Trial Register, DRKS00019086, registered on 3 January 2020. (https://www.drks.de/drks_web/navigate.do?navigationId=trial.HTML&TRIAL_ID=DRKS00019086).

## Introduction

The COVID-19 pandemic has significantly affected the quality of life of citizens all over the world. Even those who were not infected by the SARS CoV-II virus had to cope with the restrictions on personal freedom and public life [1,2]. The psychological impact of the COVID-19 pandemic has been investigated in a large number of studies worldwide and recent reviews of this research has come to the conclusion that in many countries after an initial increase in symptoms of psychological distress, people in the general population seemed to have mentally adapted to the challenges of daily life related to the COVID-19 pandemic after the first months [3–9]. Nevertheless, several studies have identified population groups which are at higher risk of becoming more severely and more durably affected [1,6,10–12] as well as social, economic and individual characteristics which make people in general more resilient against the adverse psychological effects of the pandemic [13,14].

Among other vulnerable groups, persons with severe mental illness (SMI) have been expected by many experts to become particularly affected by the psychological distress caused by fear of infection, increasing social isolation, and restricted access to mental health care services [15–27]. However, most of the studies on the impact the COVID-19 pandemic on psychopathological symptoms suggested that the majority of patients with mental illness seemed to have adapted to the challenges of the pandemic after the first wave as well as people without pre-existing mental health problems [6,16,17,28–39]. Nevertheless, these results do not indicate that the pandemic had no impact on the quality of life and the well-being of people with mental disorder at all. On the contrary, several studies indicated that whilst the number of general psychiatric inpatient admissions as well as the number of psychiatric outpatient contacts decreased significantly [11,26,32,40–45] at least some facilities reported increases of involuntary and emergency admissions due to mental disorder [24,40,41,45–48]. Overall, results are mixed; most studies provide a contradictory picture indicating not only negative effects or none whatsoever, but even positive effects of the pandemic depending on clinical and sociodemographic characteristics of the investigated target groups [12,28,31,33,39,46,48–50].

In order to develop and implement strategies to identify and support those patient groups with an increased risk of becoming negatively affected by restrictions and other challenges related to the COVID-19 pandemic, a stronger focus of research on potential resilience factors is needed [13]. Concerning other psychological and social characteristics, perceived empowerment (PE) has been identified a key resource enabling people with SMI to manage disease related adversities in the process of recovery [51–57]. PE is defined as the perception of control and mastery over one's life, and of the perceived ability to utilize one's skills to prevent or cope with life events [58–60]. PE is regarded as a crucial factor in resilience theory [60] suggesting

that PE improves an individual's capacity to cope with health related environmental challenges in a way that decreases potential negative effects on well-being and quality of life [61,62]. Only recently the improvement of PE by means of an online training program has been considered as a resource helping vulnerable population groups to cope with the challenges caused by the COVID-19 pandemic [62]. However, although the important role of PE in the process of recovery from mental disorder has been widely demonstrated, to date there is no research about the potential effects of PE as a resource that could help people with SMI to cope with the psychosocial adversities of the COVID-19 pandemic.

## Objectives

In this investigation we want to answer the following research questions:

How do patients with SMI perceive the COVID-19 pandemic having negative effects on different dimensions of their life?

How is the experience of negative COVID-19 effects associated with socioeconomic and clinical characteristics?

To what extent is change of the experience of negative effects of the COVID-19 pandemic moderated by PE and by functional impairment due to mental illness?

How is the experience of negative effects of the COVID-19 pandemic associated with the patients' subjective quality of life?

## Methods

### Design and sample

The data of this investigation were collected as part of a multi-site randomized controlled trial (RCT) for the evaluation of an intervention providing a community psychiatric service intervention (GBV) for people with SMI in addition to care as usual (CAU). Details of the study have been published in the study protocol [63]. This study was approved by the Ethics Committee of Ulm University on August 28, 2019 (application number: 259/19), and by local ethics committees (Landesärztekammer Bayern, Landesärztekammer des Saarlandes, Landesärztekammer Berlin, Landesärztekammer Sachsen, Landesärztekammer Nordrhein, Landesärztekammer Westfalen-Lippe). While the original RCT includes four follow-up assessments after baseline, only baseline and first follow-up data are used in this investigation.

Due to the COVID-19 pandemic the study onset had been postponed for 6 months from January 2020 to June 2020. In addition, the originally planned face-to-face assessment was changed to an internet based video interview with an online data collection. Only in cases where participants refused video interviewing or had not the technical requirements necessary for video interviewing, the assessment was conducted face-to-face.

Study participants were recruited at 12 locations across Germany [63]. Potential study participants were first contacted by local mental health service providers or by health insurances if they had a psychiatric diagnosis and received psychiatric treatment during the last year. After signing an informed consent all participants were screened for fulfilling the criteria of a SMI by means of a standardized screening procedure [63]. If they fulfilled the SMI criteria, participants were randomized either to GBV or to CAU and the baseline assessment was performed by a research worker.

### Assessment instruments

The "Health of the Nations Outcome Scale" (HoNOS) is an expert rating instrument to assess the impairment due to mental disorder on twelve dimensions including, agitated or aggressive

behavior, depression, hallucination, self-impairment, substance abuse, cognitive capability, social relationships, daily living, accommodation, occupation and physical health [64–66]. For this investigation the HoNOS total score was used.

The "Empowerment in Patients with Affective and Schizophrenia disorders scale" (EPAS) is a self-rating scale for the assessment of perceived empowerment at the dimensions daily living and occupation, social relationships, participation in the treatment process, sense of control and hopefulness [67,68]. Originally developed for patients with affective and schizophrenia spectrum disorders, the EPAS was found to show good psychometric properties also in use with patients with other diagnoses [67]. For this investigation the EPAS total score was used.

The brief version of the "World Health Organization Quality of Life" instrument (WHO-QOL-BREF) is a 24 item short version of the WHOQOL-100 which includes 100 items. The WHOQOL-BREF is a self-rating instrument assessing subjective quality of life at four dimensions physical health, psychological well-being, social relationships and environment and a global score including two items on perceived overall quality of life and overall satisfaction with health [69,70]. For this analysis the WHOQOL-BREF global quality of life score was used.

The "Perceived Impact of COVID-19" scale (PICOV19) was developed for the purpose of assessing how our study participants perceived the impact of the pandemic on 9 dimensions: 1. Physical well-being; 2. cognitive functioning; 3. emotional wellbeing; 4. partnership; 5. social contacts; 6. financial situation; 7. housing situation; 8. mobility; 9. future expectations. For each of these dimensions participants were asked how the COVID-19 pandemic had affected it. The generation of the PICOV19 items was based on the assumption that the effects of lockdown measures implemented by the public health authorities to reduce infection risk could be perceived positively or negatively by patients with SMI. While negative effects could be expected due to limitations of usual activities and restrictions of individual freedom, positive effects could be expected by reducing daily hassles and the need for social interaction or having more time for family activities. Therefore, the answering categories for the items of the PICOV19 scale were 1 = very negative, 2 = negative, 3 = neither positive nor negative, 4 = positive, 5 = very positive. For the purpose of data analyses all items were reversely coded into negative direction. In addition to the analyses of the single items we generated a PICOV19 sum scores of the 9 items.

## Statistical analyses

For descriptive information on sample characteristics we computed means and standard deviations or absolute and relative frequencies. The internal consistency of the PICOV-19 was tested by means of Cronbach's alpha. We also estimated means and 95% confidence intervals of the baseline and six-months follow-up (FU) assessments of the 9 dimensions and the total sum score (PICOV19) of the perceived impact of the COVID-19 pandemic on daily life.

We computed mixed effects regression models for each of the 9 dimensions of the negative effects due to the COVID-19 pandemic and for the PICOV19 total scale. We included the following independent variables: SARS-CoV-2 infection status (yes vs. no); male gender (yes vs. no), age, partnership status (having a partner vs not having a partner); education (A-levels vs lower); monthly family income (6 categories in steps of 500 €); employment status (employed vs not employed); self-reported main ICD-10 diagnosis (dummy variables using F3 as reference category); EPAS total score; HoNOS total score; time (six-months FU vs. baseline assessment) and two interaction terms for time*EPAS and time*HoNOS.

In order to analyze the impact of perceived negative effects of the COVID-19 pandemic on Quality of Life and the role of empowerment and illness severity in this process we generated a

path model linking the baseline measures of empowerment (EPAS total score) and impairment due to mental disorder (HoNOS total score) with the PICOV19 at baseline and six-months FU and with global quality of life (WHOQOL-BREF) at six-months FU. All path coefficients were controlled for sex, age, education, income, partnership status, employment status, current smoking status, body mass index (BMI) main ICD-10 diagnosis and if the patients reported a SARS-CoV II infection between baseline and six-months FU. Missing values were handled by using the full information maximum likelihood (FIML) estimator [71,72]. Overall model fit of the path model will be indicated by a non-significant Chi2 test, a Root Mean Square Error of Approximation (RMSEA) < = 0.05; a Comparative Fit Index (CFI) and a Tucker-Lewis Index (TLI) both > 0.95 and a Standardized Root Mean Square Residual (SRMR) < = 0.05 [73].

## Results

As shown in Table 1, 931 persons were included in the analyses at baseline and 768 could be included at six-months FU. The comparison of sample characteristics does not indicate any selection bias in those who were lost at follow-up.

Fig 1 shows the raw means and the 95% confidence intervals of the PICOV19 items and the PICOV19 summary scale at baseline (t0) between June 2020 and December 2020 and six-months FU (t1) between December 2020 and June 2021. At baseline, the housing situation was reported to be the least impacted with a mean of 3.09 (95% CI = 3.05–3.12), followed by partnership with a mean of 3.25 (95% CI = 3.20–3.30). The highest impact was reported for emotional well-being with a mean of 3.96 (95% CI = 3.92–4.01) as well as social contacts with a mean of 3.90 (95% CI = 3.85–3.96). The comparison of the raw means between t0 and t1 assessments indicate increased perceptions of negative effects of the COVID-19 pandemic on physical well-being (ß = 0.086; 95% CI = 0.024–0.148; p = 0.007) and on social contacts (ß = 0.081; 95% CI = 0.019–0.143; p = 0.010). All other differences were not significant as p < 0.05.

**Table 1. Sample characteristics.**

|  | Baseline<br>N = 931 | 6-months follow up<br>N = 768 | p<br>difference |
|---|---|---|---|
| Female, n (%) | 597 (64.1) | 497 (54.7) | 0.801 |
| Age, M (sd) | 42.4 (13.2) | 42.9 (13.0) | 0.362 |
| High education, n (%) | 522 (56.1) | 437 (56.9) | 0.731 |
| Living with partner, n (%) | 334 (35.9) | 278 (36.2) | 0.890 |
| Income median category | 1500–2000 € | 1500–2000 € |  |
| Employment, n (%) | 369 (39.6) | 327 (42.5) | 0.220 |
| Current smoker, n (%) | 331 (35.6) | 249 (32.4) | 0.176 |
| BMI, M (sd) | 26.4 (6.3) | 26.9 (6.3) | 0.146 |
| ICD 10 diagnosis |  |  |  |
| • F2 Psychosis, n (%) | 84 (9.0) | 65 (8.5) | 0.685 |
| • F3 Affective, n (%) | 614 (66.0) | 518 (67.5) | 0.515 |
| • F4 Anxiety, n (%) | 162 (17.4) | 129 (16.8) | 0.742 |
| • F6 Personality disorders, n (%) | 53 (5.7) | 42 (5.5) | 0.841 |
| • Other disorders, n (%) | 18 (1.9) | 14 (1.8) | 0.868 |
| SARS CoV-II Infection, n (%) | 31 (3.3) | 42 (5.5) | 0.031 |

As indicated by the distribution of main ICD-10 diagnoses at baseline 66% (n = 614) of the participants had an affective disorder, 17% (n = 162) had an anxiety disorder, 9% (n = 84) had a schizophrenia spectrum disorder, 5.7% (n = 53) had a personality disorder, and 3.1% (n = 18) had another diagnosis.

At baseline 31 (3.3%) of the participants reported that they had been positively tested for the SARS CoV-II virus. At six-months follow-up the infection rated increased significantly to 42 (5%).

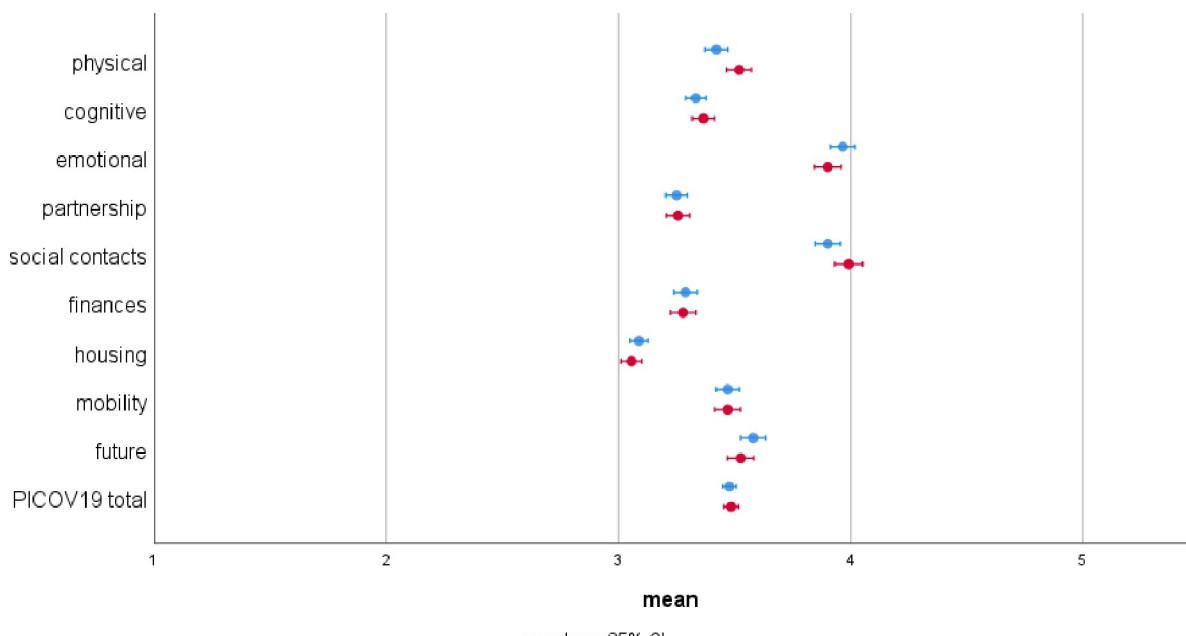

**Fig 1. Means and 95% confidence intervals of the PICOV19 items and the total score at baseline (blue) and six months follow-up (red).** The scale is reversely coded indicating: 1 = very positive effects; 2 = positive effects; 3 = neither positive nor negative effects, 4 = negative effects, 5 = very negative effects.

As indicated by final mixed effects regression models (see Table 2), the intensity of negative effects of the COVID-19 pandemic was perceived higher in participants who were positively tested of having a SARS-CoV-2 infection for the areas of physical well-being (b = 0.33; p <0.001) and mobility (ß = 0.20; p = 0.042). With increasing age participants perceived more negative effects concerning social contacts (ß = 0.0; p = 0.011), mobility (ß = 0.01; p = 0.000), and future (ß = 0.00; p = 0.004). Participants with a higher education perceived more negative effects than those with lower education with regard to cognitive capability (ß = 0.08; p = 0.027), social contacts (ß = 0.11; p = 0.023), and mobility (ß = 0.18; p = 0.000). Participants with higher income perceived more negative effects regarding emotional well-being (ß = 0.04; p = 0.009) but less negative effects regarding their financial situation (ß = -0.04; p = 0.009). Employed participants perceived more negative effects of the COVID-19 pandemic concerning their housing situation (ß = 0.06; p = 0.049) but less negative effects concerning their future than those who were not employed (ß = -0.10; p = 0.015).

Compared to participants with a diagnosis of an affective disorder, those with a schizophrenia spectrum disorder perceived less negative effects of the COVID-19 pandemic on their emotional well-being (ß = -0.30; p = 0.000), partnership (ß = -0.22; p = 0.004), social contacts (ß = -0.30; p = 0.000), their financial situation (ß = -0.26; p = 0.001), their mobility (ß = -0.18; p = 0.018) and their future (ß = -0.23; p = 0.003). Participants with a personality disorder perceived less severe negative effects than those with an affective disorder regarding their emotional well-being (ß = -0.31; p = 0.001).

Between baseline assessment and six-months FU, participants perceived increasing negative effects of the COVID-19 pandemic regarding their physical well-being (ß = 0.76; p = 0.008), their emotional well-being (ß = 0.76; p = 0.006), and their future (ß = 0.63; p = 0.027).

With increasing PE, participants perceived less negative effects of the COVID-19 pandemic regarding cognitive functioning (ß = -0.21; p = 0.000), emotional well-being (ß = -0.26;

**Table 2. Mixed effects regression models for the perceived negative effects of the COVID-19 pandemic (standardized regression coefficients and *p*-values).**

| | Dimensions of perceived negative effects of the COVID-19 pandemic | | | | | | | | | | | | | | | | |
|---|---|---|---|---|---|---|---|---|---|---|---|---|---|---|---|---|---|
| | Physical | | Cognitive | | Emotional | | Partnership | | Social contacts | | Finances | | Housing | | Mobility | | Future | |
| | ß | *p* | ß | *p* | ß | *p* | ß | *p* | ß | *p* | ß | *p* | ß | *p* | ß | *p* | ß | *p* |
| SARS-CoV-2 infection | **0.33** | **.000** | 0.16 | .065 | -0.02 | .808 | -0.07 | .436 | 0.00 | .967 | 0.04 | .675 | -0.14 | .070 | **0.20** | **.042** | 0.03 | .973 |
| Male sex | 0.03 | .538 | -0.06 | .194 | -0.05 | .311 | 0.04 | .355 | **0.10** | **.035** | -0.03 | .560 | 0.02 | .543 | 0.01 | .746 | -0.00 | .913 |
| Age | 0.00 | .100 | -0.00 | .860 | -0.00 | .363 | 0.00 | .738 | **0.00** | **.011** | -0.00 | .691 | 0.00 | .839 | **0.01** | **.000** | **0.00** | **.004** |
| Partner | -0.00 | .937 | 0.00 | .961 | 0.01 | .888 | -0.04 | .414 | 0.04 | .429 | 0.07 | .163 | **-0.10** | **.006** | -0.06 | .236 | -0.00 | .965 |
| Higher education | 0.07 | .081 | **0.08** | **.027** | 0.02 | .586 | 0.02 | .618 | **0.11** | **.013** | 0.08 | .064 | 0.06 | .075 | **0.18** | **.000** | 0.08 | .064 |
| Income | 0.01 | .639 | 0.02 | .228 | **-0.04** | **.009** | -0.01 | .628 | 0.01 | .448 | **-0.04** | **.009** | 0.00 | .586 | 0.02 | .136 | 0.01 | .700 |
| Employed | 0.01 | .759 | 0.05 | .156 | 0.05 | .280 | 0.00 | .971 | 0.05 | .305 | -0.05 | .207 | **0.06** | **.059** | 0.05 | .248 | **-0.10** | **.015** |
| ICD-10 F3 | reference | | | | | | | | | | | | | | | | | |
| ICD-10 F2 | -0.12 | .095 | -0.12 | .062 | **-0.30** | **.000** | **-0.22** | **.004** | **-0.30** | **.000** | **-0.26** | **.001** | -0.10 | .090 | **-0.18** | **.018** | **-0.23** | **.003** |
| ICD-10 F4 | -0.02 | .754 | -0.01 | .854 | -0.00 | .997 | 0.01 | .783 | -0.05 | .372 | -0.09 | .118 | 0.03 | .442 | 0.05 | .329 | 0.02 | .767 |
| ICD-10 F6 | -0.12 | .178 | -0.10 | .191 | **-0.31** | **.001** | -0.03 | .693 | -0.07 | .475 | -0.17 | .061 | 0.03 | .651 | -0.13 | .144 | 0.13 | .156 |
| ICD-10 other | -0.35 | .012 | **-0.32** | **.011** | 0.08 | .598 | 0.12 | .396 | -0.26 | .093 | 0.19 | .198 | 0.06 | .601 | -0.25 | .091 | 0.10 | .480 |
| Time | **0.76** | **.008** | 0.43 | .079 | **0.76** | **.006** | 0.35 | .205 | 0.33 | .270 | 0.08 | .770 | 0.03 | .891 | 0.43 | .126 | **0.63** | **.027** |
| EPAS total | **-0.11** | **.038** | **-0.21** | **.000** | **-0.28** | **.000** | **-0.11** | **.023** | **-0.25** | **.000** | -0.02 | .764 | **-0.09** | **.034** | -0.09 | .079 | **-0.32** | **.000** |
| Time*EPAS | **-0.18** | **.009** | **-0.12** | **.045** | **-0.21** | **.002** | -0.05 | .491 | **-0.08** | **.276** | 0.01 | .162 | -0.04 | .532 | -0.10 | .143 | **-0.17** | **.015** |
| HoNOS total | **0.02** | **.001** | 0.01 | .077 | **0.01** | **.013** | **0.01** | **.044** | 0.00 | .900 | 0.01 | .162 | **0.01** | **.012** | 0.01 | .076 | **0.01** | **.006** |
| Time*HoNOS | -0.00 | .566 | 0.00 | .936 | 0.01 | .301 | -0.01 | .089 | -0.00 | .680 | 0.00 | .734 | 0.01 | .244 | -0.01 | .268 | 0.01 | .393 |
| constant | 2.76 | .000 | 2.24 | .000 | 1.55 | .000 | 2.56 | .000 | 1.69 | .000 | 2.27 | .000 | 2.87 | .000 | 2.64 | .000 | 1.84 | .000 |
| n | 900 | | 898 | | 900 | | 885 | | 900 | | 894 | | 895 | | 898 | | 901 | |

ß = standardized regression coefficient; EPAS total = Perceived Empowerment; HoNOS = Health of the Nations Outcome Scale.

p < = 0.000), their partnership (ß = -0.11; p = 0.042), their social contacts (ß = -0.19; p = 0.001), their housing situation (ß = -0.09; p = 0.042) and their future (ß = -0.28; p = 0.000).

With increasing functional impairment due to mental illness, participants perceived increasing negative effects of the COVID-19 pandemic regarding their physical well-being (ß = 0.02; p = 0.000), emotional well-being (ß = 0.02; p = 0.003), housing situation (ß = 0.01; p = 0.029) and their future (ß = 0.01; p = 0.010).

As indicated by the coefficient for the time*EPAS interaction term, increasing PE was related to a decreasing growth of perceived negative effects of the COVID-19 pandemic between baseline assessment and six-months FU, concerning physical well-being (ß = -0.21; p = 0.004), cognitive functioning (ß = -0.13; p = 0.050), emotional well-being (ß = -0.26; p = 0.000), partnership (ß = -0.11; p = 0.042), social contacts (ß = -0.19; p = 0.001), housing situation (ß = -0.09; p = 0.042), and future (ß = -0.28; p = 0.000). The coefficients for the time-*HoNOS interaction term indicated no significant moderation effect due to participants' level of functional impairment.

Cronbach's alpha for the PICOV19 summary scale was estimated 0.75 at baseline and 0.77 at six-months FU indicating a sufficient reliability.

The results of the mixed effects regression model for the PICOV19 total score (see Table 3) indicate that having a higher education was associated with the perception of more negative effects in comparison to having a lower education (ß = 0.08; p = 0.001). Compared to participants with a diagnosis of an affective disorder, those with a diagnosis of a schizophrenia spectrum disorder (ß = -0.20; p = 0.000) and those with a diagnosis of a personality disorder (ß = -0.11; p = 0.026) experienced less negative effects of the COVID-19 pandemic. As indicated by

**Table 3. Mixed effects regression model for the PICOV19 total score (standardized regression coefficients and p values).**

| | PICOV19 total | |
|---|---|---|
| | **ß** | *p* |
| SARS-CoV-2 infection | 0.05 | .347 |
| Male sex | 0.00 | .963 |
| Age | 0.00 | .036 |
| Partner | -0.01 | .703 |
| Higher education | **0.08** | **.001** |
| Income | 0.00 | .601 |
| Employed | 0.01 | .688 |
| ICD-10 F3 | reference | |
| ICD-10 F2 | **-0.20** | **.000** |
| ICD-10 F4 | -0.01 | .669 |
| ICD-10 F6 | **-0.11** | **.026** |
| ICD-10 other | -0.13 | .094 |
| Time | **0.42** | **.002** |
| EPAS total | **-0.17** | **.000** |
| Time*EPAS | **-0.11** | **.002** |
| HoNOS total | **0.01** | **.000** |
| Time*HoNOS | 0.00 | .311 |
| Constant | 2.26 | .000 |
| n | 881 | |

the coefficients for the main effects of the HoNOS and the EPAS, baseline values of the PICOV19 total score increased with increasing level of functional impairment due to mental illness (ß = 0.01; p = 0.000), but decreased with increasing level of PE (ß = -0.17; p = 0.000). While the overall perception of negative effects increased between baseline assessment and six-months FU (ß = 0.42; p = 0.002), the increase was significantly lowered by the level of PE measured by the EPAS (ß = -0.11; p = 0.000).

As indicated by the standardized path coefficients presented in Fig 2 the functional impairment due to mental illness, measured by the HoNOS total score at baseline, was related to an increased negative impact of the pandemic six-months FU at baseline (ß = 0.157; p = 0.000) and at six-months FU (ß = 0.078; p = 0.038) and to decreased quality of life at six-months FU (ß = -0.247; p = 0.000) < 0.001). Baseline empowerment was associated with a decreased negative impact of the pandemic at baseline (ß = -0.194; p = 0.000) and at six-months FU (ß = -0.175; p = 0.000) and with increased quality of life at six-months FU (ß = 0.203; p = 0.000).

The decomposition of total direct and indirect effects revealed a total positive effect from empowerment at baseline to quality of life at a six-months FU (ß = 0.281; p = 0.000) consisting of a direct effect (ß 0.203; p = 0.000) and a total indirect effect (ß = 0.078; p = 0.000) which largely runs over the perception of a negative impact of the pandemic on the PICOV19 score six-months FU(ß = 0.059; p = 0.000) and over the combined effect of the PICOV19 score at baseline and at six-months FU(ß = -0.031; p = 0.000). The total effect of the HoNOS baseline score to quality of life at a -six-months FU(ß = 0.288; p = 0.000) consists of the direct effect (ß = - 0.247; p = 0.000) and a total indirect effect (ß = -0.042; p = 0.004) mainly running via the perception of a negative impact of the COVID-19 pandemic on the PICOV19 total score at baseline (ß = -0.026; p = 0.045) and over the combined effect of the negative impact of the pandemic at baseline and at six-months FU (ß = -0.025; p = 0.001).

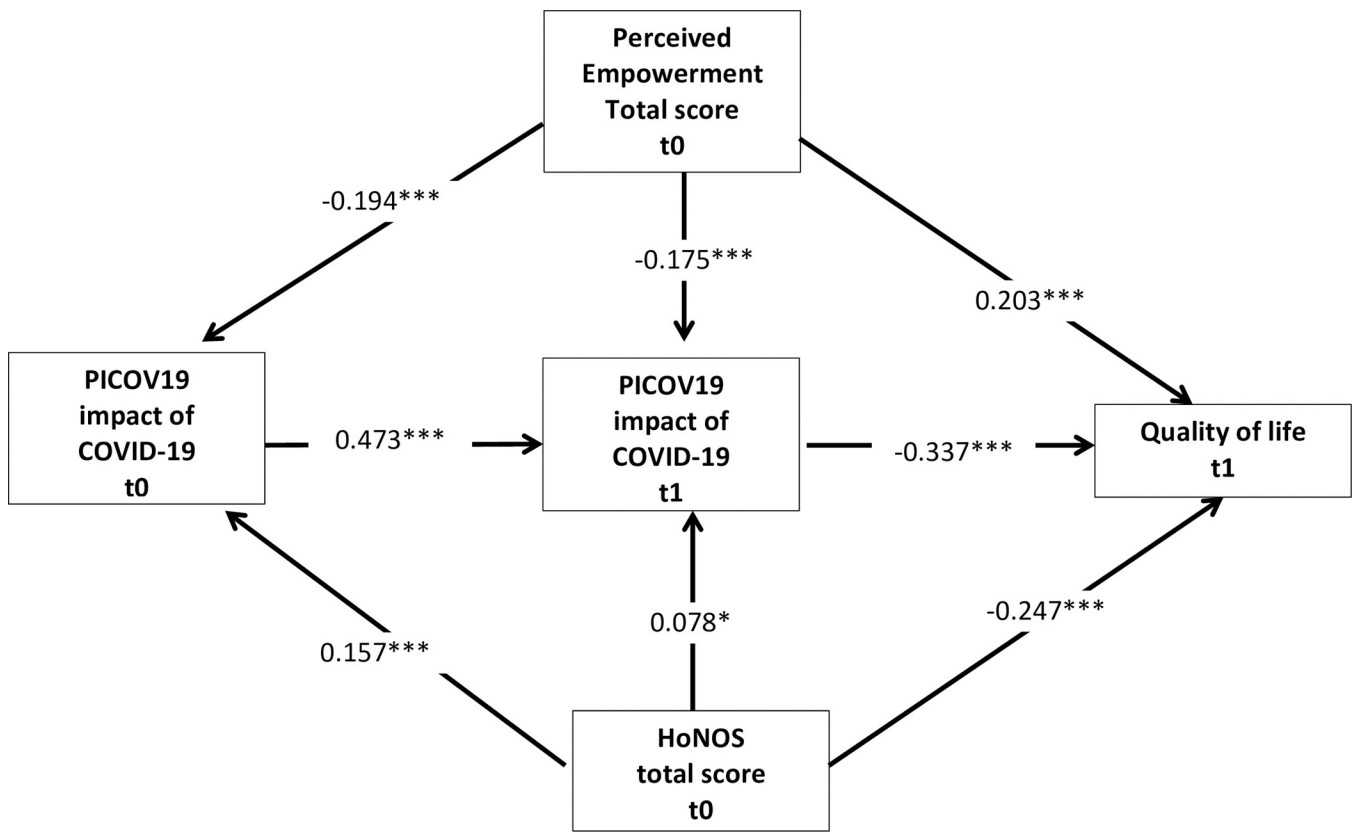

**Fig 2. Standardized path coefficients (ß); controlled for SARS-CoV-2 infection in the past, sex, age, education, income, partnership status, employment status, current smoking status, BMI and main ICD-10 diagnosis.** * $p <= 0.05$, ** $p <= 0.01$, *** $p <= 0.001$.

As indicated by the $R^2$ coefficients, the model explained 14% of the variance of the perceived negative impact of the COVID-19 pandemic measured by the PICOV19 score at baseline and 37.6% of the variance of the PICOV19 score at six-months FU and 36% of the variance of quality of life at six-months follow-up.

The overall model fit parameter reveals a very good model fit with a nonsignificant $Chi^2$ 2.567 (p = 0.2771), RMSEA of 0.018 (95% CI = 0.000–0.071) with a probability of 79% to be <= 0.05; CFI = 0.999, TLI = 0.975 and SRMR = 0.005.

## Discussion

Overall, our study results confirm the hypothesis shared by many researchers that the negative consequences of the COVID-19 pandemic on people with mental disorders go far beyond the physical health impairments potentially caused by a SARS CoV-II infection [16,18,19,21–27]. We found direct effects of a SARS CoV-II infection only for perceived negative effects on physical well-being and on mobility. This leaves the question of what affects the perception of negative effects on the other dimensions of life. Against the expectations of several authors [17,19], compared to a diagnosis of depression, a diagnosis of schizophrenia was rather related to a lower perception of negative effects on six of the nine dimensions of daily living. However, the lower perception of negative effects due to the COVID-19 pandemic in patients with schizophrenia compared to other diagnosis groups was also reported in other studies [28,33,37,50]. An explanation for these counterintuitive results could be that the restrictions in public life

following lockdown measures were associated with lower levels of social and environmental stress, while the limitations in face to face treatment contacts in many countries were substituted by digital media consultation [74–79]. Nevertheless, our results also indicate that perceived negative effects of the COVID-19 pandemic increase with increasing level of functional impairment due to mental disorders, at least on the dimensions of physical and emotional well-being, partnership, housing situation, and expectations for the future but also on the PICOV19 score. These results make it obvious that patients who are strongly affected by their mental illness also need the most intensive support in coping with the negative effects of the COVID-19 pandemic.

Our study results also suggest that PE, in the sense of a perceived control over key dimensions of life, represents an essential resource in people with SMI to cope with adversities of the COVID-19 pandemic. Increasing PE is associated with lower perception of negative effects of the COVID-19 pandemic on seven of the nine dimensions assessed with the PICOV19. Moreover, PE was associated with a reduced increase of perceived negative effects during the course of the pandemic over six months on the dimensions of physical and emotional well-being, cognitive functioning, social contacts, expectations for the future, and on the PICOV19 total score.

## Limitations

Limitations result mainly from the fact that our study design was not developed to measure effects of the COVID-19 pandemic. As a consequence we do not have real baseline measures for the effects of the pandemic. In addition, the fact that the study sample includes only participants who were eligible for the enrollment in the GBV program, limits the representability of the results with regard to the population of patients with SMI.

## Conclusions

Independent of their SARS-CoV II infection status patients with SMI had a high risk of experiencing negative consequences of the COVID-19 pandemic on their daily lives. Perception of negative consequences increased with severity of illness and decreased with the level of perceived empowerment (PE). Promotion of PE should be considered an important component of mental health care in this target group [63,67].

## Acknowledgments

We acknowledge the support of the local service providers for community-based mental health care (GBV), specifically: Netzwerk integrierte Gesundheitsversorgung (NiG) Pinel gGmbH, Berlin; Medizinisch technischen Versorgungszentrum (MtVZ) Dresden GmbH, Dresden; Gesellschaft für psychische Gesundheit in Nordrhein-Westfalen (GpG NRW) gGmbH, Solingen; Ivita Rheinland-Pfalz und Saarland gGmbH, Koblenz; INTEGRE GmbH, Augsburg; and VICENTRO gGmbH, München. Moreover, we thank all of the local health service providers who have contributed to patient recruitment. Furthermore, deep appreciation goes to the statutory health insurance, which will contribute to successful project performance through diverse supportive actions.

## Author Contributions

**Conceptualization:** Annabel Sandra Mueller-Stierlin, Thomas Becker, Reinhold Kilian.

**Data curation:** Annabel Sandra Mueller-Stierlin, Friedrich Meixner, Jutta Lehle, Anne Kohlmann, Mara Schumacher, Stefanie Woehler, Anke Haensel, Sabrina Reuter, Katrin Herder, Nicole Bias.

**Formal analysis:** Annabel Sandra Mueller-Stierlin, Reinhold Kilian.

**Funding acquisition:** Annabel Sandra Mueller-Stierlin, Thomas Becker, Reinhold Kilian.

**Investigation:** Annabel Sandra Mueller-Stierlin, Reinhold Kilian.

**Methodology:** Annabel Sandra Mueller-Stierlin, Reinhold Kilian.

**Project administration:** Annabel Sandra Mueller-Stierlin, Reinhold Kilian.

**Resources:** Annabel Sandra Mueller-Stierlin, Reinhold Kilian.

**Software:** Annabel Sandra Mueller-Stierlin, Reinhold Kilian.

**Supervision:** Annabel Sandra Mueller-Stierlin, Reinhold Kilian.

**Validation:** Annabel Sandra Mueller-Stierlin, Reinhold Kilian.

**Visualization:** Annabel Sandra Mueller-Stierlin, Reinhold Kilian.

**Writing – original draft:** Annabel Sandra Mueller-Stierlin, Reinhold Kilian.

**Writing – review & editing:** Annabel Sandra Mueller-Stierlin, Friedrich Meixner, Jutta Lehle, Anne Kohlmann, Mara Schumacher, Stefanie Woehler, Anke Haensel, Sabrina Reuter, Katrin Herder, Nicole Bias, Thomas Becker, Reinhold Kilian.

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
