## [Decision Letter · Decision Letter 0]

29 Sep 2022

Perceived empowerment and the impact of negative effects of the COVID-19 pandemic on the quality of life of persons with severe mental illness.

PONE-D-22-21801

Dear Dr. Mueller-Stierlin,

We’re pleased to inform you that your manuscript has been judged scientifically suitable for publication and will be formally accepted for publication once it meets all outstanding technical requirements.

Kind regards,

Dylan A Mordaunt, MD, MPH, FRACP

Academic Editor

PLOS ONE

Journal Requirements:

1. We note that you have indicated that data from this study are available upon request. PLOS only allows data to be available upon request if there are legal or ethical restrictions on sharing data publicly. For information on unacceptable data access restrictions, please see http://journals.plos.org/plosone/s/data-availability#loc-unacceptable-data-access-restrictions. 

Additional Editor Comments:

Thank you for your submission, which meets the criteria for publication.

Reviewers' comments:

Reviewer's Responses to Questions

**Comments to the Author**

1. Is the manuscript technically sound, and do the data support the conclusions?

Reviewer #1: Yes

2. Has the statistical analysis been performed appropriately and rigorously? 

Reviewer #1: Yes

3. Have the authors made all data underlying the findings in their manuscript fully available?

Reviewer #1: Yes

4. Is the manuscript presented in an intelligible fashion and written in standard English?

Reviewer #1: Yes

5. Review Comments to the Author

Reviewer #1: This study is a part of a larger randomized controlled trial. It reports the results of an initial interview and one follow up interview. It uses well known instruments, describes the results clearly, uses appropriate statistical methods and discuss them in a comprehensive way.

6. PLOS authors have the option to publish the peer review history of their article (what does this mean?). If published, this will include your full peer review and any attached files.

Reviewer #1: **Yes: **Adib Essali

---

## [Editor Report · Acceptance letter]

10 Oct 2022

PONE-D-22-21801 

Perceived empowerment and the impact of negative effects of the COVID-19 pandemic on the quality of life of persons with severe mental illness. 

Dear Dr. Mueller-Stierlin:

I'm pleased to inform you that your manuscript has been deemed suitable for publication in PLOS ONE. Congratulations! Your manuscript is now with our production department. 

Kind regards, 

on behalf of

Associate Professor Dylan A Mordaunt 

Academic Editor

PLOS ONE